# Association between low internal health locus of control, psychological distress and insulin resistance. An exploratory study

Maria C. M. Eriksson[1]*, Jesper Lundgren[2], Margareta Hellgren[1,3], Ying Li[4], Cecilia Björkelund[1], Ulf Lindblad[1], Bledar Daka[1]

1 Primary Health Care, School of Public Health and Community Medicine, Institute of Medicine, University of Gothenburg, Gothenburg, Sweden, 2 Department of Psychology, University of Gothenburg, Gothenburg, Sweden, 3 The Skaraborg Institute, Skövde, Sweden, 4 Biostatistics, School of Public Health and Community Medicine, University of Gothenburg, Gothenburg, Sweden

* maria.c.m.eriksson@gu.se

**Data Availability Statement:** Complete data cannot be made publicly available for ethical and legal reasons according to the Swedish regulations

## Abstract

### Aim

To assess the hypothesis that low internal health locus of control (IHLC) and psychological distress (PD) are associated with insulin resistance.

### Materials and methods

In 2002–2005, a random population sample of 2,816 men and women aged 30–74 years participated (76%) in two municipalities in southwestern Sweden. This study included 2,439 participants without previously known diabetes or cardiovascular disease. IHLC was measured by a global scale and PD was measured by the 12-item General Health Questionnaire. Insulin resistance was estimated using HOMA-ir. General linear models were used to estimate differences in HOMA-ir between groups with low IHLC, PD, and both low IHLC and PD, respectively.

### Results

Five per cent (n = 138) had both PD and low IHLC, 62 per cent of participants (n = 1509) had neither low IHLC nor PD, 18 per cent (n = 432) had PD, and 15 per cent (n = 360) low IHLC. Participants with both low IHLC and PD had significantly higher HOMA-ir than participants with neither low IHLC nor PD (Δ = 24.8%, 95%CI: 12.0–38.9), also in the fully adjusted model (Δ = 11.8%, 95%CI: 1.5–23.0). Participants with PD had significantly higher HOMA-ir (Δ = 12%, 95%CI: 5.7–18.7), but the significance was lost when BMI was included in the model (Δ = 5.3%, 95%CI:0.0–10.8). Similarly, participants with low IHLC had significantly higher HOMA-ir (Δ = 10.1%, 95%CI: 3.5–17.0) but the significance was lost in the fully adjusted model (Δ = 3.5%, 95%CI: -1.9–9.3).

of the "Act concerning the Ethical Review of Research Involving Humans (2006:460)" and the Swedish Ethical Reviews Authority. Public availability would compromise participant confidentiality or privacy. Upon request, a list of codes or meaning units can be made available after removal of details that may risk the confidentiality of the participants. To access such data, please contact the University of Gothenburg, Sahlgrenska Academy, Institute of Medicine, Department of Public Health and Community Medicine/Primary Health Care, Box 453, 40530 Gothenburg, Sweden, (generalpractice@allmed.gu.se) or Bledar Daka (bledar.daka@allmed.gu.se).

**Funding:** This work was supported by a research grant from The Local Research and Development Council Gothenburg and Södra Bohuslän; The Healthcare Committee, Region Västra Götaland; and by grants from the Swedish state under the agreement between the Swedish government and the county councils, the ALF-agreement (ALFGBG-938433 to U.L.). The funders had no role in study design, data collection and analysis, decision to publish, or preparation of the manuscript.

**Competing interests:** The authors have declared that no competing interests exist.

## Conclusions

Internal health locus of control (IHLC) and psychological distress (PD) were associated with insulin resistance. Especially individuals with both PD and low IHLC may need special attention.

## Introduction

Complex links between depression and diabetes have been increasingly revealed [1], but a greater understanding is needed both about the mechanisms in the development towards type 2 diabetes (T2D) and its prevention. T2D is a metabolic disorder, caused by insulin resistance, the inability of cells to respond adequately to insulin, together with a progressive loss of insulin secretion from the beta cells in the pancreas. Insulin resistance is strongly associated with obesity and its increase is associated with development from healthy metabolism to prediabetes to T2D. Lifestyle interventions in subjects with prediabetes can prevent or delay the onset of T2D through physical activity and healthy weight [2].

Physical activity and other health behaviours have been studied in relation to psychological constructs about outcome expectancy [3]. Locus of control is such a construct, and concerns whether outcomes are perceived as results of behaviour or personal characteristics, or else a function of chance, luck, fate, or control by others [4]. Locus of control is described as internal or external, with external in turn divided into chance and powerful others [4]. The construct of locus of control is related to social learning theory and is connected to the theory of learned helplessness and attributional style theory. Locus of control has been shown to be associated with anxiety and depression [5]. Internal locus of control has been shown to be associated with better health [6]. Low internal locus of control has been shown to be associated with later overweight and obesity [7] as well as metabolic impairment measured by HbA1c [8]. Overweight and obesity have also been linked to low internal health locus of control IHLC) [9], which is a measurement of expectancy about health outcomes specifically [10].

Comorbidity of T2D and depression is common. A bidirectional association between the two has been suggested [11], with physiological mechanisms including deregulation of the HPA-axis, low-grade inflammation and lifestyle behaviour [1]. Wider measurements of mental illness such as psychological distress (PD) have also been linked to development of T2D [12], although less studied. Moreover, stress responses have been linked to both T2D and depression, through mechanisms in the hypothalamic-pituitary-adrenalin-axis [13, 14]. To our knowledge, no studies on the association between IHLC and insulin resistance in adults have been published. Moreover, no studies have investigated whether the presence of both PD and low IHLC are associated with a higher incidence of T2D or insulin resistance.

The aim of this study was to assess the associations between low internal health locus of control (IHLC), psychological distress (PD), and insulin resistance. We also aimed to investigate the association of the presence of both low IHLC and PD with insulin resistance.

## Materials and methods

### Subjects

In 2002–2005, 2,816 men and women in the municipalities of Vara (n = 1,811) and Skövde (n = 1,005) in southwestern Sweden participated in a health survey. The participants, aged 30–75 years, were randomly sampled, with an oversampling of persons younger than 50 years of age. The participation rate was 76%. The participants had anthropometric measurements

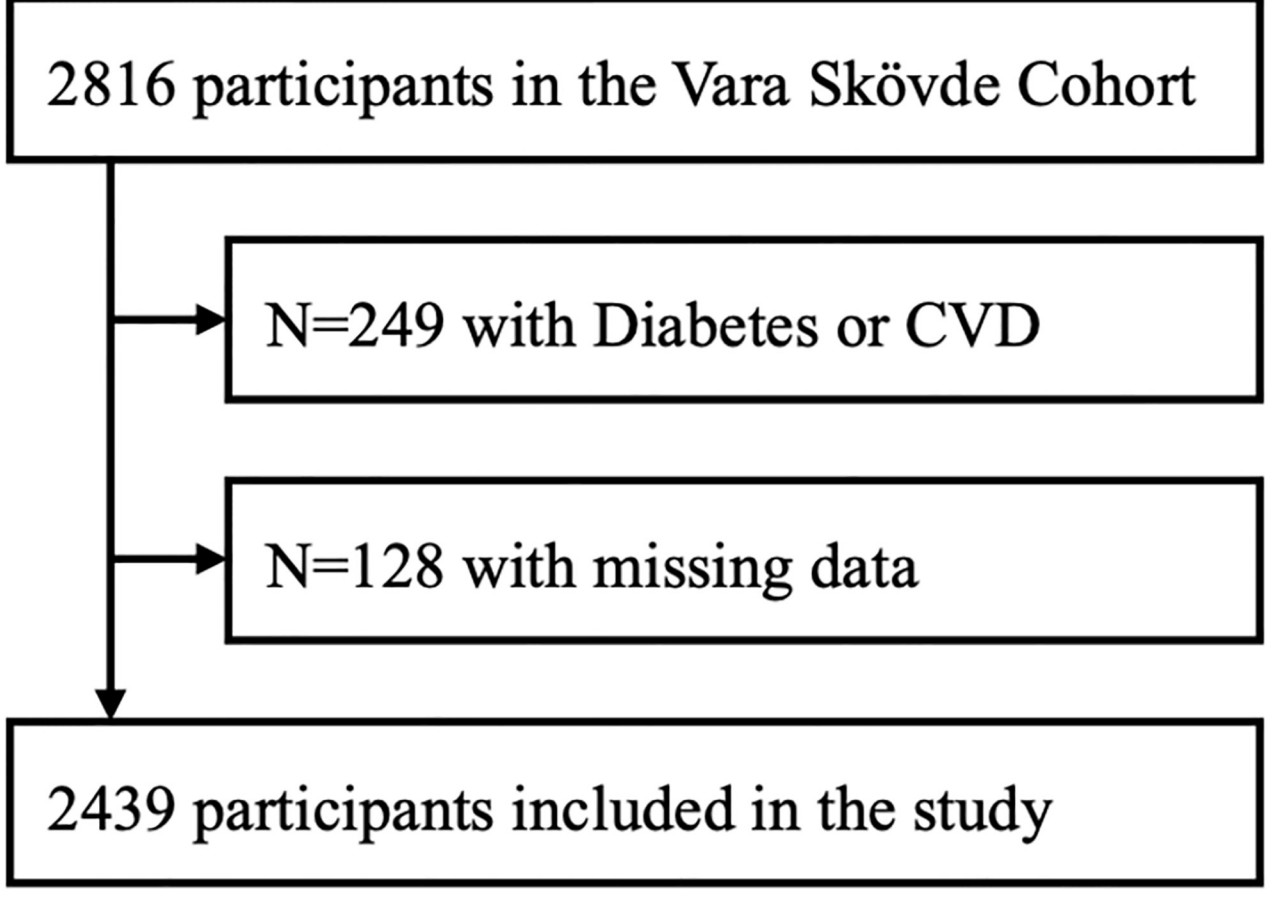

**Fig 1. Flowchart.** In all, 2,439 participants were included, after exclusion of 249 persons with Diabetes Mellitus or cardiovascular disease, and 128 persons with missing data for insulin resistance, psychological distress, or internal health locus of control.

taken, answered questionnaires and performed an oral glucose tolerance test. To avoid effects of medication in insulin resistance participants with diabetes mellitus and/or cardiovascular disease (atrial fibrillation, myocardial infarction, angina pectoris, stroke and heart failure, n = 249) were excluded. Moreover, participants with missing information for the question-naires IHLC or GHQ-12, or for insulin resistance (n = 128) were excluded, and a total of 2,439 participants remained for the present study (see flowchart in Fig 1).

## Ethics

The Regional Ethics Committee at the University of Gothenburg, Sweden, approved the study: Dnr 199–01, and Dnr 036–12. All participants signed written informed consent.

## Measurements

Specially trained nurses collected blood samples, made anthropometric measurements, and administered questionnaires to collect data on medical history, socio-demographics and life-style, as previously described in detail [15]. Insulin resistance was estimated by the homeostasis model assessment of insulin resistance, HOMA-ir, calculated by (fasting glucose x fasting insu-lin)/22.5 [16]. Psychological distress (PD) was measured by the validated 12-item General

Health Questionnaire (GHQ-12) and dichotomized with the cut-off at 12 points or more [17]. Internal health locus of control (IHLC) was measured by a global scale, previously used by Lindström and colleagues [9], and recently validated [18]. The question is phrased: "Do you believe you can do something yourself to maintain a good health", with the three response alternatives "Yes, I believe that one's own effort is very important"; "Yes, I believe that one's own effort has some importance"; and "No, I do not believe that one's own effort has any importance". IHLC was dichotomized according to the response pattern by merging the latter two and labelled high/low IHLC. PD and IHLC were then combined to indicate whether the participants had neither of the factors IHLC or PD, or both, Fig 2.

**Fig 2. Internal health locus of control and psychological distress combined into four groups.**

Educational level was assessed by a question with ten response alternatives, presented as the three groups primary school, high school, and higher education. Current smoking was defined as daily smoking (yes/no). Alcohol consumption was assessed by questions concerning the quantities of beer, wine, or spirits, respectively, consumed during the past 30 days, and presented in total grams per week. Leisure-time physical activity (LTPA) was measured by a validated questionnaire [19], and has four levels named sedentary, low, moderate, and high LTPA. Body weight was measured with participants in light clothes to the nearest 0.1 kg, standing height to the nearest centimetre, and body mass index (BMI) was calculated (weight in kg/ height in metres squared).

## Statistical analysis

Standard methods were used for descriptive statistics. HOMA-ir was log-transformed with base 10 in the analyses due to skewness in the distribution. General linear models were used to estimate differences in means in HOMA-ir between groups, adjusted for age, sex, education, alcohol consumption, daily smoking, BMI and physical activity. Due to no significant interaction between sex and the conditions (PD vs no PD, IHLC vs none) we analysed men and women together. The possible effect of physical activity on the associations between low IHLC and HOMA-ir was investigated, first with an interaction term (low IHLC * physical activity), and then in physical activity-stratified analysis. The mean differences between groups were converted into per cent of HOMA-ir with 95% confidence intervals (CI). We also considered physical activity as a mediator in the causal pathway between internal health locus of control and HOMA-ir. A mediation analysis was included to identify possible direct association of internal health locus of control on HOMA-ir, and the indirect association mediated by physical activity. All tests were two-sided and statistical significance was assumed at $p<0.05$. The statistical analyses were carried out using R version 4.2.0.

## Results

In this Swedish sample of men and women five per cent (n = 138) had both PD and low IHLC, 18 per cent (n = 432) had PD, and 15 per cent (n = 360) low IHLC, and 62 per cent (n = 1,509) reported having neither low IHLC nor PD. Median value for HOMA-ir was 1.16 (q1-q3: 0.80–1.76). Table 1 presents characteristics of the cohort. There was a moderate correlation between IHLC and PD ($r_s$ = 0.48, p = 0.017).

We investigated differences in HOMA-ir in four groups described in Fig 2. The group with neither low IHLC nor PD had lower levels of HOMA-ir than the other three groups; low IHLC; PD; and both low IHLC and PD, even when adjusting for age, sex, education, smoking and alcohol consumption, presented in Table 2. The association between HOMA-ir and the combination of both low IHLC and PD remained statistically significant also in the final model, adjusting for age, sex, education, smoking, alcohol consumption, BMI and physical activity ($\Delta$ = 11.8%, 95% CI: 1.5–23.0, p = 0.02) (Table 2). Further analyses with the group with both low IHLC and PD as reference showed that the group with both low IHLC and PD had higher insulin resistance than the group with only low IHLC and the group with only PD. However, these differences were not statistically significant after adjustments in the final model (both vs IHLC $\Delta$ = 9.6% p = 0.14; both vs PD $\Delta$ = 7.58% p = 0.26).

There was an association between PD and HOMA-ir in all models before adjusting for BMI. However, after adjusting for BMI the association was no longer significant (Table 3). Participants with low IHLC had significantly higher HOMA-ir ($\Delta$ = 10.1%, 95%CI:3.5–17.0, p<0.005). The association remained significant after adjusting for age, sex, education, smoking, alcohol consumption, and BMI ($\Delta$ = 5.8%, 95% CI: 0.3–11.7, p = 0.04). However, the

**Table 1. Characteristics of study participants in the Vara-Skövde cohort, Sweden, 2002–2005.**

| Characteristics | All | Groups based on PD and IHLC | | | |
|---|---|---|---|---|---|
| | | *None* | *High PD* | *Low IHLC* | *Both* |
| | n = 2439 | n = 1509 | n = 432 | n = 360 | n = 138 |
| Age, years, mean (SD) | 46.3 (10) | 45.8 (11) | 45.3 (10) | 48.1 (12) | 50.0 (11) |
| Sex, men, n (%) | 1205 (49) | 762 (51) | 186 (43) | 198 (55) | 59 (43) |
| Educational level n (%) | | | | | |
| *Primary school* | 584 (24) | 307 (20) | 98 (23) | 129 (36) | 50 (36) |
| *High school* | 1007 (41) | 623 (41) | 182 (42) | 144 (40) | 58 (42) |
| *Higher education* | 807 (33) | 557 (37) | 146 (34) | 79 (22) | 25 (18) |
| Occupation n (%) | | | | | |
| *Employed* | 1912 (78) | 1233 (82) | 318 (74) | 274 (76) | 87 (63) |
| *Retired* | 312 (13) | 143 (10) | 65 (15) | 62 (17) | 42 (30) |
| *Other*[a] | 136 (6) | 82 (5) | 38 (9) | 11 (3) | 5 (4) |
| BMI mean (SD) | 26.8 (5) | 26.3 (4) | 26.9 (5) | 26.7 (4) | 27.5 (5) |
| HOMA-ir, [b] | 1.2 (0.8–1.8) | 1.1 (0.8–1.7) | 1.2 (0.8–1.9) | 1.2 (0.8–1.8) | 1.5 (0.9–2.2) |
| IHLC n (%) | | | | | |
| *Yes, to a very high extent* | 1941 (80) | 1509 (100) | 432 (100) | - | - |
| *Yes, to some extent* | 493 (20) | - | - | 357 (99) | 136 (99) |
| *No, not possible* | 5 (0.2) | | | 3 (0.8) | 2 (1.4) |
| GHQ-12, mean (SD) | 9.6 (4) | 7.9 (2) | 15.1 (4) | 8.1 (2) | 14.5 (4) |
| ≥12, n (%) | 570 (23) | - | 432 (100) | - | 138 (100) |
| LTPA n (%) | | | | | |
| *Sedentary* | 163 (7) | 80 (5) | 37 (9) | 32 (9) | 14 (10) |
| *Low* | 1363 (56) | 788 (52) | 254 (59) | 227 (63) | 94 (68) |
| *Moderate* | 765 (31) | 542 (36) | 113 (26) | 86 (24) | 24 (17) |
| *High* | 78 (3) | 60 (4) | 12 (3) | 6 (1.7) | - |
| Alcohol consumption [c] | 26 (8–62) | 29 (9–63) | 25 (8–61) | 22(6–59) | 20 (0–50) |
| Daily smoking n (%) | 432 (18) | 209 (14) | 96 (22) | 83 (23) | 44 (32) |

[a] includes students, unemployed

[b] mIU/L (median, q1-q3),

[c] g/week, (median, q1-q3),

12 g alcohol is equivalent to approximately 1 glass of wine (12–15 cl) or 1 small beer (33 cl). PD: Psychological Distress, IHLC: Internal Health Locus of Control, Both: low IHLC and high PD, SD: Standard deviation, BMI: Body mass index, kg m -2, HOMA-ir: the homeostatis model assessment of insulin resistance, LTPA: leisure time physical activity.

association between low IHLC and HOMA-ir attenuated and was not significant when physical activity was included in the model (Table 3). Mediation analysis showed that 35.3% of the total association of internal health locus of control on HOMA-ir was mediated by physical activity. The interaction term of IHLC and physical activity was significant (p = 0.017). Stratified analyses showed that the association between IHLC and HOMA-ir was insignificant in individuals with high LTPA (Δ = -5.6%, 95% CI: -15.4–5.4, p = 0.31) but statistically significant in individuals with low LTPA even after adjusting for age, sex, education, smoking, alcohol consumption and BMI (Δ = 7.3%, 95% CI: 0.9–14.1, p = 0.03).

## Discussion

In this study, the presence of both low IHLC and PD was strongly and independently associated with high levels of insulin resistance, and the association remained statistically significant

**Table 2. Associations between low internal health locus of control (IHLC), Psychological distress and insulin resistance (HOMA-ir).**

| | N | Mean ref mIU/L | Mean difference | CI | p |
|---|---|---|---|---|---|
| *Crude.* | | | | | |
| None (ref.) | 1509 | **1.12** | | | |
| PD | 432 | | 10.3% (PD vs. none) | 0.7–16.1 | <0.01 |
| Low IHLC | 360 | | 8.10% (IHLC vs. None) | 3.28–17.9 | 0.03 |
| Both | 138 | | 24.8% (both vs. none) | 12.0–38.9 | <0.01 |
| *Adjusted for age and sex.* | | | | | |
| None (ref.) | 1509 | | | | |
| PD | 432 | | 11.6% (PD vs. none) | 4.5–19.2 | <0.01 |
| Low IHLC | 360 | | 6.6% (IHLC vs. None) | -0.7–14.5 | 0.08 |
| Both | 138 | | 24.4% (both vs. none) | 11.7–38.5 | <0.01 |
| *Adjusted for age, sex and education.* | | | | | |
| None (ref.) | 1487 | | | | |
| PD | 426 | | 10.7 (PD vs. none) | 3.6–18.3 | <0.01 |
| Low IHLC | 352 | | 4.6% (IHLC vs. None) | -2.6–12.4 | 0.22 |
| Both | 133 | | 22.5% (both vs. none) | 9.8–36.7 | <0.01 |
| *Adjusted for age, sex, education, smoking, alcohol and BMI.* | | | | | |
| None (ref.) | 1414 | | | | |
| PD | 405 | | 5.1% (PD vs. none) | -0.8–11.33 | 0.09 |
| Low IHLC | 341 | | 4.2 (IHLC vs. None) | -2.1–10.8 | 0.20 |
| Both | 129 | | 15.2% (both vs. none) | 4.8–26.6 | <0.01 |
| *Adjusted for age, sex, education, smoking, alcohol, BMI and physical activity.* | | | | | |
| None (ref.) | 1396 | | | | |
| PD | 396 | | 3.9% (PD vs. none) | -2.0–10.1 | 0.20 |
| Low IHLC | 337 | | 1.9% (IHLC vs. None) | -4.3–8.5 | 0.55 |
| Both | 125 | | 11.8% (both vs. none) | 1.54–23.0 | 0.02 |

All estimates are based on the general linear model, presented with 95% confidence interval (CI) and p-value (p).

in the fully adjusted model. Psychological distress (PD) was strongly associated with insulin resistance and the association was almost significant in the final model. There was a strong and significant association between low internal health locus of control (IHLC) and insulin resistance. The association attenuated after adjustments for physical activity. Interestingly, differences in the association between IHLC and insulin resistance were observed based on the level of physical activity. Low IHLC was associated with higher insulin resistance only in subjects with low level of leisure time physical activity suggesting a central role of LTPA in the association between IHLC and insulin resistance.

No previous study investigating the association between IHLC and insulin resistance in adults has been conducted to our knowledge. We know of only one previous study in which an association between higher HbA1c and low internal locus of control was shown [8]. In that study, internal locus of control and socioeconomic status covaried, and the authors suggested that socioeconomic status explained the association. In our study however, the statistically significant association between low IHLC and HOMA-ir remained after adjustment for socioeconomic status, using educational level as a proxy for socioeconomic status. Another study has found that 10-year-olds with high internal locus of control had a reduced risk of overweight and obesity 20 years later [7], which would be in line with the results in the current study considering the strong link between obesity and insulin resistance.

**Table 3. Associations of psychological distress (PD) and low internal health locus of control (IHLC) with insulin resistance (HOMA-ir).**

| | N | Mean ref | MD | CI | p |
|---|---|---|---|---|---|
| **Psychological Distress** | | | | | |
| Crude | | | | | |
| | 2439 | **1.12** | 12% | 5.7–18.7 | <0.01 |
| Adjusted for age and sex | | | | | |
| | 2439 | | 13.2% | 6.8–19.9 | <0.01 |
| Adjusted for age, sex and education | | | | | |
| | 2398 | | 12.4% | 6.0–19.1 | <0.01 |
| Adjusted for age, sex, education, smoking, alcohol, and physical activity. | | | | | |
| | 2256 | | 6.5% | 1.3–12.1 | 0.01 |
| Adjusted for age, sex, education, smoking, alcohol, physical activity and BMI. | | | | | |
| | 2254 | | 5.3% | 0.0–10.8 | 0.05 |
| **Low internal health locus of control** | | | | | |
| Crude | | | | | |
| | 2439 | **1.12** | 10.1% | 3.5–17.0 | <0.01 |
| Adjusted for age and sex | | | | | |
| | 2439 | | 8.6% | 2.2–15.4 | 0.01 |
| Adjusted for age, sex and education | | | | | |
| | 2398 | | 6.8% | 0.3–13.6 | 0.04 |
| Adjusted for age, sex, education, smoking, alcohol and BMI. | | | | | |
| | 2289 | | 5.8% | 0.3–11.7 | 0.04 |
| Adjusted for age, sex, education, smoking, alcohol, BMI and physical activity. | | | | | |
| | 2254 | | 3.5% | -1.9–9.3 | 0.21 |

All estimates are based on the general linear model with $\log_{10}$HOMA-ir as the response, PD or IHLC as the variable of interest, adjusted for covariates as specified. Means and mean differences (MD) in HOMA-ir between persons with and without (reference group) psychological distress, respectively, and between persons with high (reference group) and low internal health locus of control respectively, presented with 95% confidence interval (CI) and p-value (p).

Ravaja et al. investigated the effects of interactions between locus of control and life events on serum insulin and found that the effects of the interaction were generally in the opposite direction compared with the effect of locus of control, and that the interaction effects were dependent on which type of life event had occurred [20]. In young women with internal locus of control, changes in educational/working activities, change of residence, or setting up a family was associated with high levels of serum insulin and high BMI while no association was found in young women with external locus of control [20]. The differences seem contrasting compared with our cohort. However, the discussion about the age-appropriateness of some of the life events measured in this young cohort is relevant. The adolescents in the study were between 15 and 21 years old at the first examination [20]. The extent to which 15-year-olds can choose occupation, where to live, when to move etc. is often small, which would imply low controllability of such life events. Experiments have found that adults with high internal locus of control had increased stress responses in low controllability conditions [21] and a reduced stress response when they believed in controllability in a high controllability condition [22]. Furthermore, the ability to distinguish between realistically controllable and uncontrollable events and knowing when to let go of things is beneficial [6]. Changes in coping strategies towards flexible adjustments of goals and shifting focus from trying to change the environment to changing oneself have been shown in older ages [6] and may also be relevant in younger ages. Previous findings about locus of control and age indicate that internal locus of control generally increases in young adulthood [6].

Ravaja et al. [20] did not assess perceived controllability or positive and negative emotional responses to life events, which may explain some of the variation. In our study of adults aged 30–75 years, we measured psychological distress including strain, worry and depressive symptoms that may be associated with a spectrum of stress responses of different intensity. Reactions to life events may be positive or negative, while psychological distress (GHQ-12≥12p) perhaps in most cases would be labelled as negative but potentially subclinical.

In our study, physical activity attenuated the association between IHLC and HOMA-ir and the associations varied depending on level of physical activity suggesting a central role of physical activity in mediating the possible effect of IHLC on HOMA-ir. This is consistent with previous studies that have shown that men and women with high internal locus of control were more physically active [23] and that higher sense of control was related to higher likelihood of frequent physical activity [24]. Physical activity has in turn been observed to have a dose-response relationship with insulin resistance [25]. Furthermore, the association between psychological distress and HOMA-ir found in this study is in line with the findings of increased risk of prediabetes and T2D 8–10 years later in patients with PD [12].

To our knowledge, no other studies have investigated the presence of both PD and low IHLC in insulin resistance. Our hypothesis about an association between the presence of both low IHLC and PD and higher HOMA-ir was confirmed. The moderate correlation between IHLC and PD in our study was expected and in line with previous studies on the association between locus of control and PD [7] and depression and anxiety symptoms [5]. A recent study on the relationships between locus of control, positive and negative life events, depression, and anxiety over a nine-year period showed that externality (measured by a 5-item mastery scale) predicted depressive symptoms and anxiety, and that depressive symptoms and negative life events predicted externality [26]. The GHQ-12 questionnaire used in the current study has items about depressive symptoms, like feelings of worthlessness, feeling unhappy and depressed, and items about worry and feeling under strain. A Swedish study has validated the GHQ-12 against clinical assessment of depressive disorders (including minor depression) suggesting that the GHQ-12 captures depressive disorders well [17].

In addition to previously suggested physiological mechanisms [1], there is also the lifestyle factor of behaviours common in mental illness, such as poor diet and low level of physical activity, simultaneously being risk factors for metabolic impairment and T2D. A conceptual model of reciprocal, cyclic associations based on cognitive behavioural theory has been suggested [27]. In the model, modified to the variables in our study, health outcomes like psychological distress influence IHLC, which in turn influences mediating behaviours like physical activity, the stress response and insulin resistance, and possible future health outcomes, such as T2D and depression.

## Strengths and limitations

Firstly, there may be minor selection bias concerning non-participating subjects in the Vara-Skövde Cohort (VSC). VSC is a randomly sampled cohort from two municipalities with high participation rate. However, participation has been found to be greater among healthier individuals and persons interested in the topic [28], suggesting that the present study may have fewer participants with low IHLC, thus increasing the risk for type 2 error. Secondly, due to the cross-sectional design, the present study cannot determine possible causality between IHLC, PD and insulin resistance. Thirdly, while euglycemic glucose clamp technique is gold standard for measurement of insulin resistance, HOMA-ir was used in this study. HOMA-ir is

a validated and common method in epidemiological studies. Likewise, we used validated, but self-rated and subjective measurements for leisure time physical activity and PD, which may have lower precision compared with objective measurements of physical activities and professional diagnostic interviews. Not having information on diet in this study may be a limitation since diet is strongly linked to body weight management and diabetes prevention. Finally, we used a global scale to measure IHLC. Many different scales have been used in the literature, self-efficacy perhaps being the most closely related construct. Studies on health locus of control and health behaviour have shown some contradictory findings [29], and Wallston who introduced the Multidimensional Health Locus of Control Scale (MHLC) later shifted focus from health locus of control to health self-efficacy and placed IHLC as moderator between health self-efficacy and health behaviour [30]. The global scale used in this study was recently validated and correlated with both the MHLC, and the General Self-Efficacy scale [18].

## Conclusions

The presence of both low internal health locus of control and psychological distress in combination was strongly and independently associated with insulin resistance, also when adjusting for BMI and physical activity. Individuals with both low internal health locus of control and psychological distress may need special attention from primary care. In individuals with low levels of leisure time physical activities, interventions to improve internal health locus of control might decrease insulin resistance and consequently the risk for type 2 diabetes.

## Acknowledgments

The authors would like to thank all participants from Vara and Skövde who made this study possible and the team of nurses that worked hard to collect the data.

## Author Contributions

**Conceptualization:** Maria C. M. Eriksson, Ulf Lindblad, Bledar Daka.

**Data curation:** Maria C. M. Eriksson, Ulf Lindblad.

**Formal analysis:** Maria C. M. Eriksson, Ying Li.

**Investigation:** Margareta Hellgren, Ulf Lindblad, Bledar Daka.

**Methodology:** Maria C. M. Eriksson, Jesper Lundgren, Ying Li, Ulf Lindblad, Bledar Daka.

**Supervision:** Jesper Lundgren, Cecilia Björkelund, Ulf Lindblad, Bledar Daka.

**Writing – original draft:** Maria C. M. Eriksson, Bledar Daka.

**Writing – review & editing:** Maria C. M. Eriksson, Jesper Lundgren, Margareta Hellgren, Ying Li, Cecilia Björkelund, Ulf Lindblad, Bledar Daka.

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
