## [Decision Letter · Decision Letter 0]

1 Mar 2023

PONE-D-23-01333Association between low internal health locus of control, psychological distress and insulin resistance, a cross-sectional study

PLOS ONE

Dear Dr. Eriksson,

Thank you for submitting your manuscript to PLOS ONE. After careful consideration, we feel that it has merit but does not fully meet PLOS ONE’s publication criteria as it currently stands. Therefore, we invite you to submit a revised version of the manuscript that addresses the points raised during the review process.

Please submit your revised manuscript by Apr 15 2023 11:59PM, If you will need more time than this to complete your revisions, please reply to this message or contact the journal office at plosone@plos.org. Please include the following items when submitting your revised manuscript:A rebuttal letter that responds to each point raised by the academic editor and reviewer(s). You should upload this letter as a separate file labeled 'Response to Reviewers'.A marked-up copy of your manuscript that highlights changes made to the original version. You should upload this as a separate file labeled 'Revised Manuscript with Track Changes'.An unmarked version of your revised paper without tracked changes. You should upload this as a separate file labeled 'Manuscript'.

We look forward to receiving your revised manuscript.

Kind regards,

Victor Manuel Mendoza-Nuñez, PhD

Academic Editor

PLOS ONE

Additional Editor comments:

Please, in addition to the corrections in your manuscript, you must respond point by point to the comments of the reviewers.

Another additional comment to that of the reviewers is the need to specify in the title that it is an exproratory study, the following title is suggested: "Association between low internal health locus of control, psychological distress and insulin resistance. An exploratory study"

Reviewer 1

This cross-sectional study examines the relationship between insulin resistance, internal locus of control, and psychological stress in a random sample of individuals from southwestern Sweden. The method and results sections are adequately presented.

Every table should be self-explanatory. Tables 2 and 3 have imprecise titles (to the extent that the footnotes of the two tables indicate which is the outcome variable and which are the adjustment variables, which is unnecessary). A suggestion would be:  Table 2. Association of insulin resistance (HOMA-ir) with psychological distress (PD) and low internal locus of control (IHLC). Similarly, the description of the transformation of estimates to percentages is unnecessary because it has already been covered in the section on data analysis.

Even though the study is exploratory, it would be helpful to provide more evidence for the hypotheses about the physiological mechanisms behind the links found in the study.

Reviewer 2

The results presented are interesting, however, the authors report that only 5% of the population they studied met the criteria that decided to associate low internal health locus of control and phychological distress with insulin resistance (HOMA-ir).

Therefore, they decided to add other variables to the mathematical model that they proposed, such as physical activity in a stratified manner, which was not contemplated at the beginning.

The authors decided to add variable by variable to the mathematical model until it stopped giving statistically significant results, so they stopped focusing their article on the initial proposal, to assess the hypothesis that low internal health locus of control (IHLC) and psychological Distress (PD) are associated with insulin resistance.

I believe that the results must be ordered and focus on the initial proposal.

Reviewers' comments:

Reviewer's Responses to Questions

**Comments to the Author**

1. Is the manuscript technically sound, and do the data support the conclusions?

Reviewer #1: Yes

Reviewer #2: Partly

2. Has the statistical analysis been performed appropriately and rigorously? 

Reviewer #1: Yes

Reviewer #2: Yes

3. Have the authors made all data underlying the findings in their manuscript fully available?

Reviewer #1: No

Reviewer #2: Yes

4. Is the manuscript presented in an intelligible fashion and written in standard English?

Reviewer #1: Yes

Reviewer #2: Yes

5. Review Comments to the Author

Reviewer #1: This cross-sectional study examines the relationship between insulin resistance, internal locus of control, and psychological stress in a random sample of individuals from southwestern Sweden. The method and results sections are adequately presented.

Every table should be self-explanatory. Tables 2 and 3 have imprecise titles (to the extent that the footnotes of the two tables indicate which is the outcome variable and which are the adjustment variables, which is unnecessary). Una sugerencia sería: Table 2. Association of insulin resistance (HOMA-ir) with psychological distress (PD) and low internal locus of control (IHLC). Similarly, the description of the transformation of estimates to percentages is unnecessary because it has already been covered in the section on data analysis.

Even though the study is exploratory, it would be helpful to provide more evidence for the hypotheses about the physiological mechanisms behind the links found in the study.

Reviewer #2: The results presented are interesting, however, the authors report that only 5% of the population they studied met the criteria that decided to associate low internal health locus of control and phychological distress with insulin resistance (HOMA-ir).

Therefore, they decided to add other variables to the mathematical model that they proposed, such as physical activity in a stratified manner, which was not contemplated at the beginning.

The authors decided to add variable by variable to the mathematical model until it stopped giving statistically significant results, so they stopped focusing their article on the initial proposal, to assess the hypothesis that low internal health locus of control (IHLC) and psychological Distress (PD) are associated with insulin resistance.

I believe that the results must be ordered and focus on the initial proposal.

6. PLOS authors have the option to publish the peer review history of their article (what does this mean?). If published, this will include your full peer review and any attached files.

Reviewer #1: No

Reviewer #2: No

---

## [Author Response · Author response to Decision Letter 0]

15 Apr 2023

 Answer: We have made changes to meet the style requirements. 

 Answer: In Methods, in the ethics statement, row 112, we have written: All participants signed written informed consent.

 Answer: We have added to the Data Avaliablity statement: 

Complete data cannot be made publicly available for ethical and legal reasons according to the Swedish regulations of the “Act concerning the Ethical Review of Research Involving Humans (2006:460)” and the Swedish Ethical Reviews Authority. Public availability would compromise participant confidentiality or privacy. Upon request, a list of codes or meaning units can be made available after removal of details that may risk the confidentiality of the participants. To access such data, please contact the University of Gothenburg, Sahlgrenska Academy, Institute of Medicine, Department of Public Health and Community Medicine/Primary Health Care, Box 453, 40530 Gothenburg, Sweden, (generalpractice@allmed.gu.se) or Bledar Daka (bledar.daka@allmed.gu.se).

Additional Editor comments:

Please, in addition to the corrections in your manuscript, you must respond point by point to the comments of the reviewers.

Another additional comment to that of the reviewers is the need to specify in the title that it is an exproratory study, the following title is suggested: "Association between low internal health locus of control, psychological distress and insulin resistance. An exploratory study"

 Answer: Thank you, we agree and have changed the title accordingly.

Reviewer 1

This cross-sectional study examines the relationship between insulin resistance, internal locus of control, and psychological stress in a random sample of individuals from southwestern Sweden. The method and results sections are adequately presented.

Every table should be self-explanatory. Tables 2 and 3 have imprecise titles (to the extent that the footnotes of the two tables indicate which is the outcome variable and which are the adjustment variables, which is unnecessary). A suggestion would be: Table 2. Association of insulin resistance (HOMA-ir) with psychological distress (PD) and low internal locus of control (IHLC). Similarly, the description of the transformation of estimates to percentages is unnecessary because it has already been covered in the section on data analysis.

 Answer: Thank you, we have changed the titles of the table 2 and 3 and the footnotes. The title of table is now “Associations of psychological distress (PD) and low internal health locus of control (IHLC) with insulin resistance (HOMA-ir)”. We decided to write insulin resistance last in the title since that is the outcome variable. We have also removed the description of the transformation of estimates to percentages. 

Even though the study is exploratory, it would be helpful to provide more evidence for the hypotheses about the physiological mechanisms behind the links found in the study.

 Answer: We have added some text and a reference on the hypothesis about the physiological mechanisms. Row 79-81, and row 334-337.

Reviewer 2

The results presented are interesting, however, the authors report that only 5% of the population they studied met the criteria that decided to associate low internal health locus of control and phychological distress with insulin resistance (HOMA-ir).

Therefore, they decided to add other variables to the mathematical model that they proposed, such as physical activity in a stratified manner, which was not contemplated at the beginning.

The authors decided to add variable by variable to the mathematical model until it stopped giving statistically significant results, so they stopped focusing their article on the initial proposal, to assess the hypothesis that low internal health locus of control (IHLC) and psychological Distress (PD) are associated with insulin resistance.

I believe that the results must be ordered and focus on the initial proposal.

 Answer: Thank you for drawing our intention to our unclear description! We have changed the order of the results, in the results (row 159-162, and row 169-254), discussion (row 257-266) and abstract (row 35-48). We added confounders s based on theoretical models to avoid confounding. Knowledge from previous studies on IHLC and physical activity contributed to our decision to continue with the analyses with physical activity.

---

## [Decision Letter · Decision Letter 1]

7 May 2023

Association between low internal health locus of control, psychological distress and insulin resistance. An exploratory study.

PONE-D-23-01333R1

Dear Dr. Maria Eriksson

We’re pleased to inform you that your manuscript has been judged scientifically suitable for publication and will be formally accepted for publication once it meets all outstanding technical requirements.

Kind regards,

Victor Manuel Mendoza-Nuñez, PhD

Academic Editor

PLOS ONE

---

## [Editor Report · Acceptance letter]

12 May 2023

PONE-D-23-01333R1 

Association between low internal health locus of control, psychological distress and insulin resistance. An exploratory study. 

Dear Dr. Eriksson:

I'm pleased to inform you that your manuscript has been deemed suitable for publication in PLOS ONE. Congratulations! Your manuscript is now with our production department. 

Kind regards, 

on behalf of

Dr. Victor Manuel Mendoza-Nuñez 

Academic Editor

PLOS ONE